

# Phenethyl isothiocyanate inhibits the carcinogenic properties of hepatocellular carcinoma Huh7.5.1 cells by activating MAPK/PI3K-Akt/p53 signaling pathways

Jiao Du[1,*], Yuting Zhang[1,*], Jiajia Chen[1], Libo Jin[2], Liying Pan[1], Pengyu Lei[1] and Sue Lin[1]

[1] College of Life and Environmental Science, Wenzhou University, Wenzhou, Zhejiang, China
[2] Institute of Life Sciences, Wenzhou University, Wenzhou, Zhejiang, China
[*] These authors contributed equally to this work.

Corresponding author
Sue Lin, iamkari@163.com

## ABSTRACT

**Background**. Hepatocellular carcinoma (HCC) is an aggressive malignancy with limited effective treatment options. Phenethyl isothiocyanate (PEITC) is a bioactive substance present primarily in the cruciferous vegetables. PEITC has exhibited anti-cancer properties in various cancers, including lung, bile duct, and prostate cancers. It has been demonstrated that PEITC can inhibit the proliferation, invasion, and metastasis of SK-Hep1 cells, while effectively inducing apoptosis and cell cycle arrest in HepG2 cells. However, knowledge of its anti-carcinogenic effects on Huh7.5.1 cells and its underlying mechanism remains elusive. In the present study, we aim to evaluate the anti-carcinogenic effects of PEITC on human HCC Huh7.5.1 cells.

**Methods**. MTT assay and colony formation assay was performed to investigate the anti-proliferative effects of PEITC against Huh7.5.1 cells. The pro-apoptosis effects of PEITC were determined by Annexin V-FITC/PI double staining assay by flow cytometry (FCM), mitochondrial transmembrane potential (MMP) measurement, and Caspase-3 activity detection. A DAPI staining and terminal deoxynucleotidyl transferase-mediated dUTP nick-end labeling (TUNEL) assay was conducted to estimate the DNA damage in Huh7.5.1 cells induced by PEITC. Cell cycle progression was determined by FCM. Transwell invasion assay and wound healing migration assay were performed to investigate the impact of PEITC on the migration and invasion of Huh7.5.1 cells. In addition, transcriptome sequencing and gene set enrichment analysis (GSEA) were used to explore the potential molecular mechanisms of the inhibitory effects of PEITC on HCC. Quantitative real-time PCR (qRT-PCR) analysis was performed to verify the transcriptome data.

**Results**. MTT assay showed that treatment of Huh7.5.1 cells with PEITC resulted in a dose-dependent decrease in viability, and colony formation assay further confirmed its anti-proliferative effect. Furthermore, we found that PEITC could induce mitochondrial-related apoptotic responses, including a decrease of mitochondrial transmembrane potential, activation of Caspase-3 activity, and generation of intracellular reactive oxygen species. It was also observed that PEITC caused DNA damage and cell cycle arrest in the S-phase in Huh7.5.1 cells. In addition, the inhibitory effect of PEITC on the migration and invasion ability of Huh7.5.1 cells was assessed. Transcriptome sequencing analysis further suggested that PEITC could activate the typical MAPK,

PI3K-Akt, and p53 signaling pathways, revealing the potential mechanism of PEITC in inhibiting the carcinogenic properties of Huh7.5.1 cells.

**Conclusion**. PEITC exhibits anti-carcinogenic activities against human HCC Huh7.5.1 cells by activating MAPK/PI3K-Akt/p53 signaling pathways. Our results suggest that PEITC may be useful for the anti-HCC treatment.

## INTRODUCTION

According to the latest global cancer statistics in 2020, primary liver cancer is the sixth most frequently diagnosed cancer and the third leading cause of cancer-related death worldwide (*Siegel et al., 2021*; *Sung et al., 2021*). Primary liver cancer can be categorized into three major histological subtypes, namely hepatocellular carcinoma (HCC), intrahepatic cholangiocarcinoma (ICC), and combined HCC-ICC. Among them, HCC is the most prevalent form of primary liver cancer accounting for ~90% of cases and is a severe neoplastic disease that poses a serious threat to human health (*Sia et al., 2017*). It is an aggressive tumor that usually develops in the context of chronic liver disease and cirrhosis, with the poor prognosis due to its occult onset and high recurrence rate (*Hartke, Johnson & Ghabril, 2017*). For individuals diagnosed with early HCC, surgical resection, liver transplantation, and radiofrequency ablation are currently recognized as the three primary radical treatments (*Fan et al., 2023*). Those with advanced HCC have the options to choose from targeted drug therapy, chemical immunotherapy, and radiotherapy, among which targeted therapy is a prevalent area of exploration in advanced HCC clinical research (*Chakraborty & Sarkar, 2022*). Despite its potential, targeted medication has limitations in targeting and therapeutic efficiency, along with the emergence of acquired resistance (*Llovet et al., 2008*; *Sato et al., 2020*; *Chen et al., 2020*; *Ma et al., 2021*; *Zhang et al., 2022*; *Fan et al., 2023*). Hence, exploring new chemotherapy with high bioavailability, substantial anti-cancer efficacy, and minimal adverse effects is a crucial area of focus in the management of HCC.

Natural compounds are currently receiving increasing attention as one of the most feasible solutions for hard-to-treat cancers (*Molinski et al., 2009*; *Fontana et al., 2020*; *Kubczak, Szustka & Rogalińska, 2021*; *Tewari et al., 2022*). Phenethyl isothiocyanate (PEITC), a cancer-fighting substance, is abundant in various cruciferous vegetables such as mustard, cabbage and turnip (*Fowke, 2007*; *Cheung & Kong, 2010*). Numerous studies have demonstrated the inhibitory effects of PEITC on multiple cancers involving the lung, the bile duct and the prostate (*Hecht, 1996*; *Tusskorn et al., 2013*; *Zhang et al., 2016*; *Soundararajan & Kim, 2018*). PEITC could significantly inhibit the proliferation of both human breast cancer cells MCF-7 and prostate cancer cells PCa (*Morris & Dave, 2014*; *Zhang et al., 2016*). A possible association between the inhibitory role of PEITC on lung cancer cells and AKT phosphorylation has been reported, revealing that PEITC could curb

the activation of the PI3K/AKT pathway and ultimately regulate its downstream effectors to trigger apoptosis (*Kassie et al., 2010*; *Zhang et al., 2017*). PEITC has also been considered to impede the migration and invasion of SKOV3 and HO8910 ovarian cancer cells as well as AGS gastric cancer cells *via* inhibiting mTOR-STAT3, MAPK, NF-κB, and other signaling pathways (*Yang et al., 2010*). In terms of cell cycle arrest, PEITC was found to induce cycle arrest in prostate cancer cells through inhibiting cycle-dependent protein kinase (*Wang et al., 2008*). Moreover, it was revealed that PEITC could induce apoptosis in cancer cells, such as porcine jejunal epithelial cells IPEC-J2, by enhancing cellular oxidative stress (*Trachootham, Alexandre & Huang, 2009*; *Gorrini, Harris & Mak, 2013*). On the one hand, PEITC increased the accumulation of reactive oxygen species (ROS) to play its anti-tumor effects (*Jutooru et al., 2014*); on the other hand, it bound covalently with glutathione (GSH) and resulted in decreased intracellular GSH levels, ultimately leading to apoptosis (*Zhang & Talalay, 1998*). In addition, as a broad-spectrum anti-cancer natural product, PEITC combined with other chemotherapy drugs has been clarified to greatly increase the anti-tumour efficacy (*Cang et al., 2014*). For example, a combined treatment with paclitaxel (PTX) has been proved to be an effective inhibition against breast cancer, and with PARP inhibitors to enhance the sensitivity of ovarian cancer (*Cang et al., 2014*; *Jia et al., 2021*). Currently, active research is being conducted to reveal the inhibitory effects of PEITC on the carcinogenic activity of diverse cancer cells. However, there is rather scarce data regarding the influence of PEITC on HCC. The underlying mechanism by which PEITC affects the activity of HCC has not been fully elucidated.

Accumulating evidence has revealed that the inhibitory effects of natural compounds were diverse in accordance with the types of HCC cells. Previously, the anti-cancer properties of PEITC in HCC cell lines SK-Hep1 and HepG2 has been demonstrated (*Rose et al., 2003*; *Hwang & Lee, 2006*; *Pocasap, Weerapreeyakul & Thumanu, 2018*). Among them, HepG2 cells, with the retention of various liver-specific metabolic functions, are usually adopted for studying liver metabolism, toxicity of xenobiotics, and liver disease mechanisms, whereas Sk-Hep1 cells are commonly utilized for the study of HCC invasion and metastasis (*Wu et al., 2019*; *Stanley & Wolf, 2022*). Its has been found that PEITC can inhibit the proliferation, invasion, migration, and metastasis of SK-Hep1 cells, while effectively inducing apoptosis, cell cycle arrest, and ROS generation in HepG2 cells (*Rose et al., 2003*; *Hwang & Lee, 2006*; *Pocasap, Weerapreeyakul & Thumanu, 2018*). The Huh7.5.1 cell line presents unique advantages in the research of hepatitis C virus (HCV)-related HCC, making it an ideal model (*Kawamoto et al., 2020*). However, knowledge of the anti-carcinogenic effects of PEITC on Huh7.5.1 cells and its underlying mechanism remains elusive.

The present study aims to investigate the effects of PEITC on proliferation, invasion and migration, oxidative stress, apoptosis, and cell cycle distribution in Huh7.5.1 cells. Our findings showed that PEITC can significantly suppress the proliferative effect and the migration and invasion ability, enhance the intracellular oxidative stress, induce cell cycle arrest, and activate the apoptotic pathways in Huh7.5.1 cells. Furthermore, transcriptome sequencing was performed to clarify the potential mechanism of PEITC action through

the activation of typical MAPK, PI3K-Akt and p53 signaling pathways. The present results demonstrated that PEITC could be a promising new anti-HCC biotherapeutic.

## MATERIALS & METHODS

### Cell culture and reagents

The human HCC Huh7.5.1 cells were procured from the China Typical Culture Collection Center (CCTCC) and cultured in Dulbecco's Modified Eagle Medium (DMEM) (catalog no. C11995500BT; Gibco, Waltham, MA, USA) supplemented with 10% fetal bovine serum (FBS) (catalog no. 04-001-1ACS; Biological Industries, Beit Haemek, Israel) and 1% v/v penicillin-streptomycin (catalog no. P1400; Solarbio, Beijing, P. R. China). The Huh7.5.1 cells used in this study were within the range of passages 5 to 10. Cells were incubated at 37 °C with a 5% $CO_2$ concentration.

Purified PEITC (>98%) was purchased from Aladdin (catalog no. 2257-09-2; Merck, Rahway, NJ, USA) and dissolved in dimethylsulphoxide (DMSO) (catalog no. D8371; Solarbio, Beijing, P. R. China) to prepare a 500 mmol/L stock solution, which was then stored at −80 °C.

### MTT assay and colony formation assay

MTT assay was used to determine the anti-proliferative effect of PEITC on Huh7.5.1 cells. Huh7.5.1 cells were seeded in triplicate in 96-well plates ($3 \times 10^3$ cells per well) and incubated overnight. After treatment with varying concentrations of PEITC for 48 h, cells in each well were incubated with 30 µL of MTT (catalog no. 298-93-1; Solarbio) at 37 °C for 3.5 h. Once incubation was complete, the supernatant was discarded and 150 µL of DMSO (catalog no. D103272; Aladdin, Shanghai, China) was added into each well to fully dissolve the formazan crystal. To determine the absorbance at the wavelength of 490 nm in individual wells, a microplate spectrometer (SPARK, Tecan, Swiss) was used. The half maximal inhibitory concentration ($IC_{50}$) values were calculated using GraphPad Prism 9.0 software. The cell survival rate was calculated using the following formula: cell survival rate (%) = (the average OD value of the treatment group/the average OD value of the control group) ×100%.

In the colony formation assay, Huh7.5.1 cells were introduced into a 6-well plate ($5 \times 10^3$ per well), incubated overnight, and then treated with either 15 or 30 µmol/L PEITC or 0.1% DMSO control for a duration of 12 h. Afterwards, cells were co-incubated within fresh medium and monitored for seven days. Then, the cell colonies were washed three times with phosphate buffer solution (PBS) (catalog no. C20012500BT; Gibco) and fixed with 4% paraformaldehyde (catalog no. BL539A; Biosharp, Heifei, P. R. China) for 15 min at room temperature. Following fixation, the cell colonies were further washed three times with PBS and stained with crystal violet (catalog no. C0121; Beyotime, Haimen, P. R. China) for 15 min. The stained colonies were counted and analyzed using ImageJ and GraphPad Prism 9.0.

### Transwell invasion assay and wound healing migration assay

For the transwell invasion assay, 600 µL of serum-free medium was added to the lower chamber in a sterile 24-well plate. The transwell chamber was then carefully placed into

the lower chamber with tweezers. Subsequently, Huh7.5.1 cells were inoculated into the transwell chamber ($2 \times 10^4$ cells per well) and incubated overnight. After co-culture with 15 and 30 µmol/L PEITC or 0.1% DMSO for 48 h, the cells adhering to the lower surface were washed with PBS and fixed for 15 min with 4% paraformaldehyde at room temperature followed by staining with crystal violet for 15 min. The migrated cells were photographed and counted with a bright field microscopy (TS2-S-SM; Nikon, Tokyo, Japan).

For the wound healing migration assay, Huh7.5.1 cells were inoculated into 6-well plates ($1 \times 10^5$ cells per well). When the cells in each well reached 95%–100% confluence, a sterilized 10 µL pipette tip was used to create a straight scratch through the confluent monolayer. After rinsing with PBS, cells were treated with 15 µmol/L PEITC and 0.1% DMSO for 24 h, respectively, and photographed with a bright field microscope (TS2-S-SM; Nikon). At 0, 24, 48, 72, 96 and 120 h, the wound closure was analyzed using ImageJ software.

## Apoptosis assay by flow cytometry (FCM)

The Annexin V-FITC/PI apoptosis detection kit (BD, the United States, catalog no. 556547) was used to detect cell apoptosis, according to the manufacturer's instructions. Briefly, Huh7.5.1 cells were harvested and washed three times with PBS after incubation with 15 and 30 µmol/L PEITC or 0.1% DMSO for 12 h, respectively. Cells were then suspended with 400 µL of binding buffer followed by staining with 5 µL of fluorescein isothiocyanate-coupled annexin V (FITC) and 5 µL of propidium iodide (PI) solution. After incubation in the dark for 15 min at room temperature, samples were analyzed by FCM (NovoCyte, Agilent, the United States).

## Terminal deoxynucleotidyl transferase-mediated dUTP nick-end labeling (TUNEL) assay

A one step TUNEL apoptosis assay kit (catalog no. C1088; Beyotime) in combination with DAPI staining was used to detect PEITC-induced DNA fragmentation in Huh7.5.1 cells, according to the manufacturer's instructions. Briefly, Huh7.5.1 cells were harvested and washed three times with PBS after incubation with 15 and 30 µmol/L PEITC or 0.1% DMSO for 12 h, respectively. Subsequently, cells were fixed with 4% paraformaldehyde for 10 min and then incubated with TUNEL reaction buffer for 30 min in the dark at 37 °C after infiltration with Triton X-100 (Beyotime, P. R. China, catalog no. P0096) for 5 min. After washing three times with PBS, 10 µL of 5 mg/L DAPI solution (catalog no. E607303; BBI, Shanghai, China) was added to stain the cells in the dark for 5 min at room temperature. Samples were observed under a fluorescence microscopy (TS2-S-SM, Nikon) to view the green fluorescence of apoptotic cells at 570 nm and blue DAPI-stained nuclei at 460 nm.

## Detection of Caspase-3 activity

Huh7.5.1 cells were cultured overnight in 6-well plates and incubated with 15 and 30 µmol/L PEITC or 0.1% DMSO control for 12 h, respectively. The enzymatic activity level of Caspase-3 was assessed by identifying the cleavage of $p$-nitroaniline ($p$NA) through Caspase-3-specific substrates using a Caspase-3 assay kit (catalog no. C1116; Beyotime),

according to the manufacturer's instructions. Briefly, cells were harvested, centrifuged, and subsequently lysed in a 4 °C ice bath for a duration of 15 min. The Caspase-3 enzymatic reaction was conducted for 2 h at 37 °C, with 50 µL of cell lysate, 40 µL of reaction buffer, and 10 µL of Caspase-3 colorimetric substrate (acetyl-Asp-Glu-Val-Asp p-nitroanilide, Ac-DEVD-$p$NA) in each reaction. A microplate spectrometer (SPARK, Tecan, Swiss) was used to measure the colour intensity of $p$NA at the wavelength of 405 nm.

### Determination of oxidative stress

Huh7.5.1 cells were cultured overnight in 6-well plates and incubated with 15 and 30 µmol/L PEITC or 0.1% DMSO control for 12 h, respectively. The intracellular ROS levels were determined using the oxidation-sensitive fluorescent probe DCFH-DA (Beyotime, P. R. China, catalog no. S0033S), according to the manufacturer's instructions.

### Mitochondrial transmembrane potential (MMP) measurement

According to the manufacturer's instructions, MMP changes were measured using a Mitochondrial Membrane Potential Assay Kit with JC-1 (catalog no. C2006; Beyotime). Briefly, PEITC-treated cells were collected, washed twice with cold PBS, and then suspended with the mixture of culture medium and JC-1 staining solution (v/v =1:1). After incubation at 37 °C in the dark for 20 min, cells were centrifuged followed by tracking with 1 ×JC-1 dye. The stained cells were further detected by FCM and visualized under a fluorescence microscopy (OLYMPUS, IX73, Japan).

### Cell cycle analysis

The cell cycle distribution was determined by FCM analysis. Briefly, Huh7.5.1 cells were cultured overnight in 6-well plates and incubated with 15 and 30 µmol/L PEITC or 0.1% DMSO control for 12 h, respectively. Both detached and adhered cells were collected into a centrifuge tube and washed twice with PBS. After fixation with 75% ethanol, cells were rinsed with PBS and then incubated in 0.5 mL of PI/RNaser (BD, the United States, catalog no. 550825) at room temperature for 15 min followed by FCM analysis.

### RNA sequencing and data analysis

A total of 1 µg of RNA was extracted using Trizol (Invitrogen, Waltham, MA, USA) from Huh7.5.1 cells subjected to treatment with 15 µmol/L PEITC or 0.1% DMSO for 10 h, respectively, each with three biological replicates. Total RNA was enriched with polyA+ using Dynabeads Oligo (dT) magnetic beads (catalog no. 25-61005; Thermo Fisher Scientific, Waltham, MA, USA) and served as the input for preparing libraries with the NEBNext® Magnesium RNA Fragmentation Module (catalog no. E6150S; NEB, Ipswich, MA, USA). Subsequently, Illumina Novaseq™ 6000 was used by LC-Bio Technology Co., Ltd. (Hangzhou, China) for paired-end sequencing. Fragments per kilobase of transcript per million mapped (FPKM) reads indicated the abundance of gene expression. The statistical power of RNA sequencing was calculated in RnaSeqSampleSize (https://cqs-vumc.shinyapps.io/rnaseqsamplesizeweb/). To annotate gene function, gene set enrichment analysis (GSEA) was performed using hallmark gene sets and Kyoto Encyclopedia of Genes and Genomes (KEGG) gene sets from MSigDB.

## RNA extraction, cDNA synthesis, and quantitative real-time PCR (qRT-PCR)

Huh7.5.1 cells were seeded into 6-well plates ($1 \times 10^5$ cells per well). After overnight culture, Huh7.5.1 cells were treated with 15 µmol/L PEITC or 0.1% DMSO for 10 h, respectively, each with three biological replicates. A total of 1 µg of RNA was extracted using Trizol and then treated with DNase I. The RNA amount and purity of each sample was quantified using NanoDrop ND-1000 (NanoDrop, Wilmington, DE, USA). The values of A260/A280 of all RNA samples were between 1.8 and 2.0. The RNA integrity was assessed by Bioanalyzer 2100 (Agilent, Santa Clara, CA, USA) with RIN number > 7.0, and confirmed by electrophoresis with denaturing agarose gel. RNA was reverse-transcribed to cDNA using the MonScript™ RTIII Super Mix with dsDNase (Two-Step) (catalog no. MR05201M; Monad, Wuhan, China). The cDNA synthesis reaction was carried out at 50 °C for 15 min, followed by thermal inactivation at 85 °C for 5 min. Subsequently, the produced cDNA was utilized for qRT-PCR.

The qRT-PCR reaction was set up with a total volume of 20 µL, consisting of 10 µL SYBR Green Master Mix (catalog no. MQ10101; Monad), 0.4 µL each of forward and reverse primers, 1.5 µL cDNA template, and 7.7 µL ddH$_2$O. qRT-PCR was conducted on a BioRad CFX Connet qRT-PCR Dectction system (Bio-Rad, Hercules, CA, USA). Briefly, the reaction conditions comprised an initial pre-denaturation at 95 °C for 30 s, succeeded by 40 cycles of denaturation at 95 °C for 10 s and annealing/extension at 60 °C for 30 s. The specificity of the PCR products was confirmed through melt curve analysis, with amplification lengths ranging from 110 to 300 bp. For normalization purposes, expression of $\beta$-actin was analyzed. Three biological replicates and three technical replicates were performed. Data were collected and analyzed as previously described in *Guan et al. (2024)*. Specifically, the relative gene expression levels were determined by using the $2^{-\Delta\Delta Ct}$ method, wherein the PCR efficiencies were >95% and the R2 was >0.990. All primers used are listed in Table S1.

## Statistical analysis

All experiments were performed in three biological replicates in the present study. GraphPad Prism version 9.0 was used to assess statistical differences between groups using *t*-test or one-way ANOVA. *P*-values <0.05 were considered statistically significant.

# RESULTS

## PEITC suppresses the activity and proliferation of Huh7.5.1 cells

To determine the anti-proliferative effects of PEITC (Fig. 1A) against HCC, Huh7.5.1 cells were treated with various concentrations (0~100 µmol/L) of PEITC for 48 h and then subjected to the MTT assay. We found that the anti-proliferative effects on Huh7.5.1 cells were positively correlated to the dose of PEITC and the IC$_{50}$ value of PEITC was 29.6 µmol/L (Fig. 1B). Based on the results of the MTT assay, the 2D-clonony-formation assay was further conducted. The results showed that the number of colonies significantly decreased in the PEITC-treated Huh7.5.1 cells in a dose-dependent manner relative to the

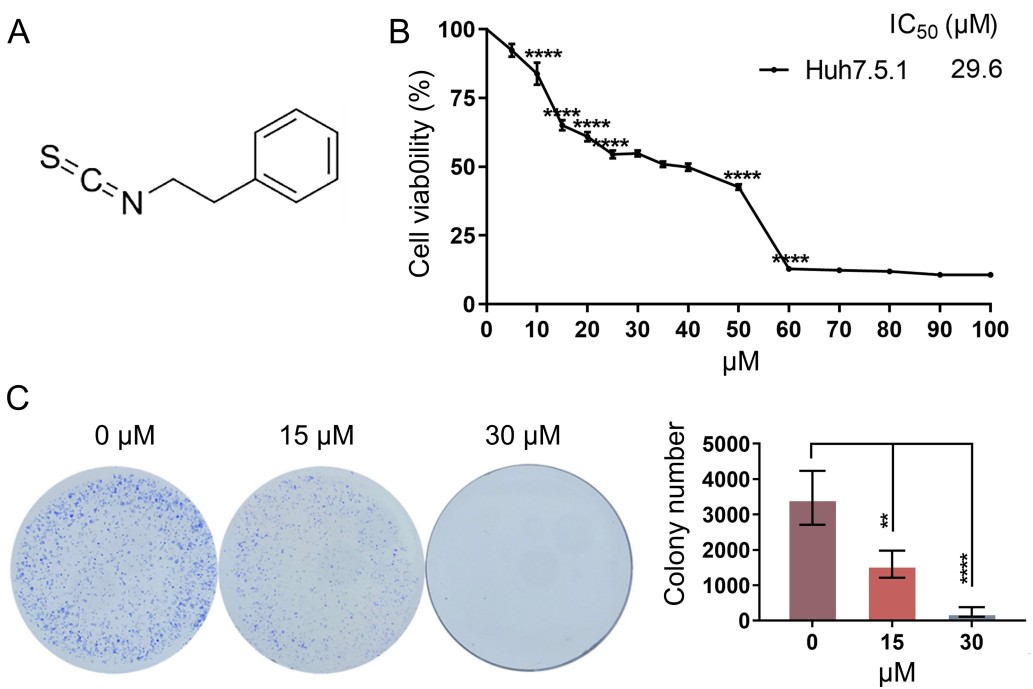

**Figure 1** **PEITC inhibits the viability and proliferation of Huh7.5.1 cells.** (A) The chemical structure formula of PEITC. (B) MTT assay of Huh7.5.1 cells treated with different concentrations (5~100 μmol/L) of PEITC and DMSO for 48 h, respectively ($n = 3$). Based on the MTT assay, the half maximal inhibiting concentration ($IC_{50}$) value was calculated ($n = 3$). (C) 2D-colony-formation assay of Huh7.5.1 cells treated with 15 and 30 μmol/L PEITC or DMSO ($n = 3$). Data were shown as mean ± SD (standard deviation). **$p < 0.01$ and ****$p < 0.0001$.

control (DMSO-treated) cells (Fig. 1C). Taken together, these results suggest that PEITC exerts anti-cancer effects on human HCC Huh7.5.1 cells.

## PEITC induces apoptosis-related cell death in Huh7.5.1 cells

In order to evaluate whether PEITC could inhibit the growth of Huh7.5.1 cells by inducing the apoptosis-related cell death, an Annexin V-FITC/PI double staining assay by FCM was conducted. The apoptotic rates of Huh7.5.1 cells significantly increased from 2.15% to 37.84% and 74.05% after treatment with 15 and 30 μmol/L PEITC for 12 h, respectively, compared with that of control cells (Fig. 2A), suggesting a significant induction of apoptosis in Huh7.5.1 cells by PEITC. As the decline of MMP is one of the landmark events in the early stage of apoptosis (*Gottlieb et al., 2003*), the change of MMP was assessed using the JC-1 probe. The results of fluorescence assay showed a increase in the green JC-1 monomer fluorescence intensity and simultaneously a progressive decline of the red JC-1 aggregate signal in the PEITC-treated Huh7.5.1 cells, indicating that PEITC markedly decreased the MMP in Huh7.5.1 cells in a dose-dependent manner (Figs. 2B and 2C). To further verify the apoptosis induced by PEITC treatment in Huh7.5.1 cells, the detection of Caspase-3 activity was also performed. The result revealed that the activity of Caspase-3 in the PEITC-treated Huh7.5.1 cells increased significantly in a dose-dependent manner,

Peerj

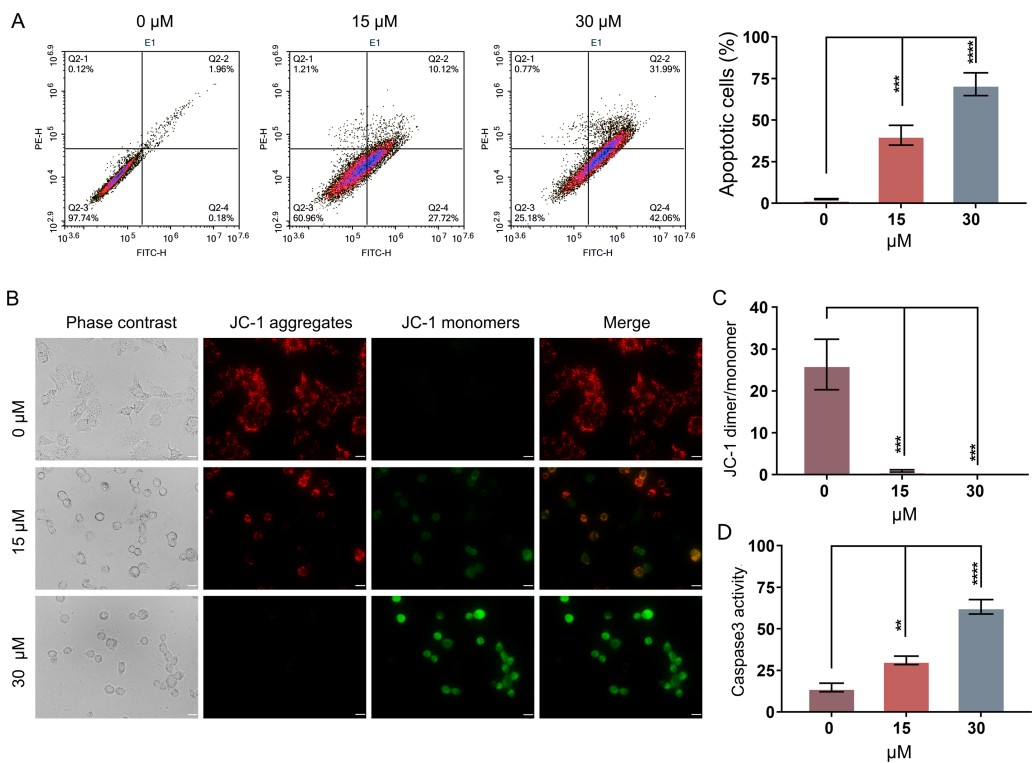

**Figure 2  PEITC induces mitochondrial-related apoptosis of Huh7.5.1 cells.** (A) Flow cytometry analysis of apoptosis by Annexin V-FITC/PI double-staining assay in Huh7.5.1 cells treated with 15 and 30 μmol/L PEITC or DMSO for 12 h. The percentage of apoptotic cells was shown on right panel ($n = 3$). (B) Mitochondrial membrane potential detection in the PEITC-treated Huh7.5.1 cells by JC-1 based fluorescence assay. Huh7.5.1 cells were incubated for 12 h with 15 and 30 μmol/L PEITC or DMSO, respectively. The red fluorescence indicates the aggregation signal of JC-1, while green indicates the monomer signal of JC-1. Scale bars = 20 μm. (C) The ratio of JC-1 aggregate/monomer in (B) ($n = 3$). (D) Determination of Caspase-3 activity in Huh7.5.1 cells incubated with 15 and 30 μmol/L PEITC or DMSO for 12 h ($n = 3$). Data were shown as mean ± SD. $^{**}p < 0.01$, $^{***}p < 0.001$ and $^{****}p < 0.0001$.

compared with that in control cells (Fig. 2D). Overall, all these results suggest that PEITC induces mitochondrial-related apoptosis in Huh7.5.1 cells.

## PEITC enhances oxidative stress and induces DNA damage in Huh7.5.1 cells

Mitochondria-related apoptotic responses, such as the decrease of MMP that promotes the enhancement of mitochondrial out membrane permeabilization (MOMP), activate the generation of ROS (*Estaquier et al., 2012*; *Liu et al., 2018*). The result of FCM analysis showed that treatment with PEITC for 12 h could significantly enhance the intracellular ROS level in Huh7.5.1 cells compared with that in control cells (Fig. 3A).

DNA damage is frequently observed in apoptotic cells (*Wu & Crowe, 2020*). When apoptosis occurs, DNA breakage, nuclear chromatin agglutination, and nuclear membrane rupture are usually induced (*CortésGutiérrez et al., 2020*). ROS accumulation has been implicated in causing DNA damage and apoptosis in diverse cancers (*Barzilai & Yamamoto,*
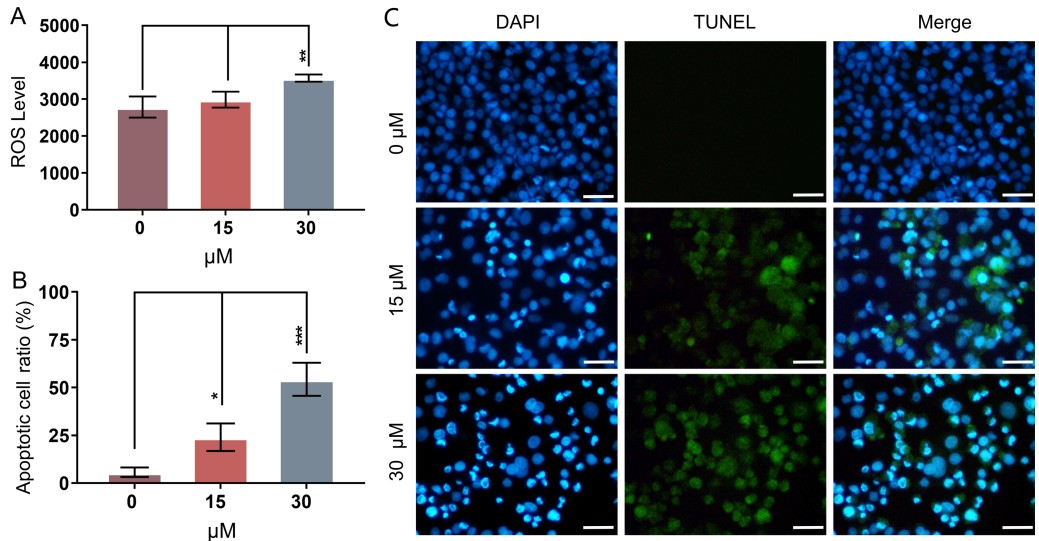

**Figure 3** **PEITC enhances oxidative stress and causes DNA damage in Huh7.5.1 cells.** (A) The levels of intracellular reactive oxygen species (ROS) were determined in Huh7.5.1 cells by flow cytometry analysis after incubation with 15 and 30 $\mu$mol/L PEITC or DMSO control for 12 h, respectively ($n = 3$). (B) The histogram represented the percentage of the apoptotic cells with DNA-damaged nuclei in three independent experiments ($n = 3$). (C) DAPI staining and TUNEL staining assay of Huh7.5.1 cells incubated with 15 and 30 $\mu$M PEITC or DMSO control for 12 h. Blue and green fluorescent signal showed nuclei and DNA fragments, respectively. DNA-damaged nuclei were revealed in the merged images. Scale bars = 100 $\mu$m. Data are shown as mean $\pm$ SD. *$p < 0.05$, **$p < 0.01$ and ***$p < 0.001$.

*2004*; *Guachalla & Rudolph, 2010*; *Guo et al., 2014*; *Sadiq, 2023*). To investigate whether PEITC could induce DNA damage in Huh7.5.1 cells, a DAPI staining and TUNEL assay was conducted. The results showed that PEITC induced a significant increase of green fluorescence in Huh7.5.1 cells in a dose-dependent manner (Figs. 3B and 3C), indicating that DNA damage occurred in the nuclei. Moreover, DAPI staining also showed a large number of nuclear ruptures (Figs. 3B and 3C). The merged images revealed the DNA-damaged nuclei in Huh7.5.1 cells (Figs. 3B and 3C).

## PEITC induces cell cycle arrest in S phase in Huh7.5.1 cells

As DNA damage usually arrests or blocks the cell cycle, we further evaluated the effect of PEITC on cell cycle distribution in Huh7.5.1 cells by FCM analysis. The results demonstrated that PEITC treatment significantly increased the population of Huh7.5.1 cells in S phase, but markedly reduced the population of cells in G2/M and G0/G1 phases, indicating that PEITC could induce S-phase arrest to play its inhibitory roles in Huh7.5.1 cells (Fig. 4).

## PEITC inhibits the migration and invasion of Huh7.5.1 cells

In order to further investigate whether PEITC could impact the migration of Huh7.5.1 cells, wound healing migration assay was conducted. The percentages of the open wound area at 0 h, 24 h, 48 h, 72 h, 96 h, and 120 h after the treatment with 15 $\mu$mol/L PEITC were calculated. Our result indicated that the migration of Huh7.5.1 cells was markedly

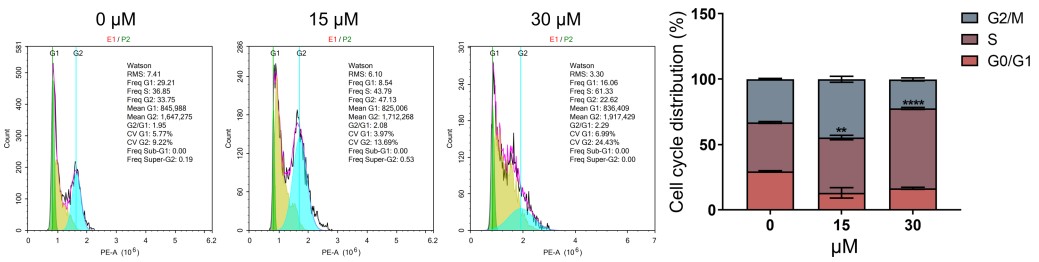

**Figure 4  PEITC induces cell cycle arrest in S phase in Huh7.5.1 cells.** The cell cycle distribution was determined in Huh7.5.1 cells incubated with 15 and 30 μmol/L PEITC or DMSO for 12 h by flow cytometry analysis. The percentage of the G0/G1, S, G2/M phases was evaluated and shown on right panel ($n = 3$). Data were shown as mean ± SD. **$p < 0.01$ and ****$p < 0.0001$.

delayed, compared with that of control cells (Fig. 5A). Specifically, at 24 h, the wound closure in the control group reached 21.7%, whereas that in the PEITC-treated group was only 2%; by 120 h, the would closure in the control group achieved 73.6%, while that in the PEITC-treated group was at 33.5%. This data clearly demonstrated the significant effect of PEITC in inhibiting cell migration (Fig. 5A). Consistent with the result of wound healing migration assay, the transwell invasion assay also confirmed that PEITC treatment had a negative influence on the invasion ability of Huh7.5.1 cells in a dose-dependent manner (Fig. 5B). Compared with the control group (2,058 invading cells), the number of invading cells in the 15 μmol/L PEITC-treated group dropped to 900, and further reduced to 130 in the 30 μmol/L PEITC-treated group (Fig. 5B). These results not only confirmed the inhibitory effect of PEITC on the invasive capacity of cells, but also presented a distinct dose-dependent pattern.

## Transcriptome analysis suggests that PEITC activates MAPK, PI3K-Akt and p53 signaling pathways in Huh7.5.1 cells

In order to explore the potential molecular mechanisms of the inhibitory effects of PEITC on HCC, transcriptome sequencing was conducted in Huh7.5.1 cells treated with or without PEITC for 10 h. The statistical power of this experimental design calculated in RnaSeqSampleSize is 0.80. Pairwise comparisons showed that a total of 2889 genes were significantly differentially expressed with a |log$_2$ fold change (FC)| ≥ 1 and a $p$-value <0.05, among which 2339 genes were significantly up-regulated and 550 down-regulated in the PEITC-treated Huh7.5.1 cells relative to control cells (Fig. 6A and Table S2). To cross-check the accuracy of the RNA-seq results, qRT-PCR analysis was conducted on eight differentially expressed genes (DEGs), including four genes involved in cell proliferation, invasion and migration (*FOS*, *PDE6G*, *RGS16* and *HBEGF*) (*MildeLangosch, 2005*; *Nikolova et al., 2010*; *Carper et al., 2014*; *Bakiri et al., 2017*; *Hu et al., 2020*; *Ji & Wang, 2023*; *Zhang et al., 2023*) and four genes related to oxidative stress, DNA damage and apoptosis (*ATF3*, *DDIT4*, *SOCS1* and *MXD1*) (*Hai et al., 1999*; *Kim, Kim & Lee, 2020*; *Ding et al., 2021*; *Ryu et al., 2022*; *Xiong et al., 2022*; *Shao et al., 2023*) (Fig. 6D). The strong consistency between the qRT-PCR validation and RNA-seq results indicated high reliability of the transcriptomic profiling data.

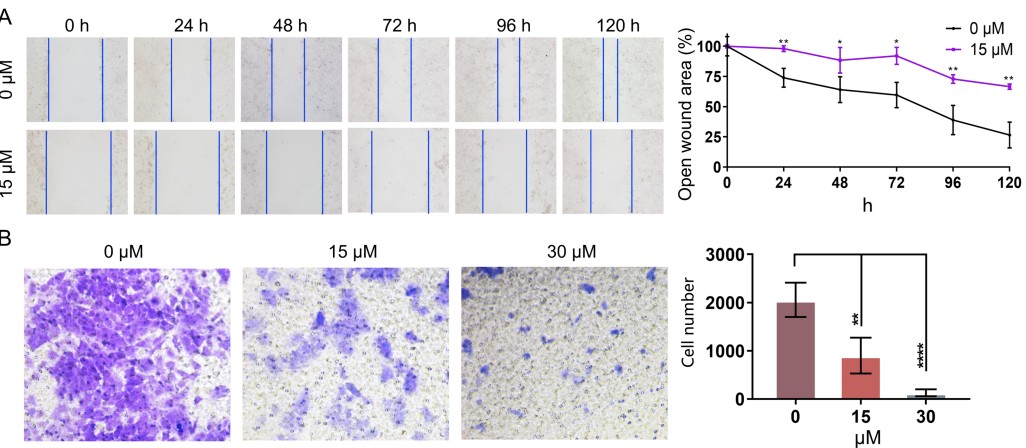

**Figure 5** **PEITC inhibits the migration and invasion of Huh7.5.1 cells.** (A) Wound healing migration assay of Huh7.5.1 cells treated with 15 μmol/L PEITC and DMSO, respectively. The relative open wound area was evaluated and quantified ($n = 3$). (B) Transwell invasion assay of Huh7.5.1 cells treated with 15 and 30 μmol/L PEITC or DMSO for 24 h. The migrated cells were stained with crystal violet and calculated by ImageJ software ($n = 3$). Data were shown as mean ± SD. $^*p < 0.05$, $^{**}p < 0.01$ and $^{****}p < 0.0001$.

The KEGG pathway analysis of DEGs manifested that the top 20 significantly enriched pathways involved two typical signaling pathways in cancer pathogenesis, namely MAPK and PI3K-Akt signaling pathways (Fig. 6B). There were 27 and 25 DEGs enriched in the MAPK and PI3K-Akt signaling pathways, respectively (Table S3). In addition, GSEA was also applied to analyze the global changes of the signaling pathways regulated by PEITC. It was found that 21 activated gene sets (NES >1.5, $p$-val <0.05, and false discovery rate (FDR) < 0.25) were identified in Huh7.5.1 cells after PEITC treatment (Table 1), including hallmarks E2F targets, PI3K-Akt-mTOR signaling and p53 pathway (Fig. 6C). The core enrichment genes for these three hallmark gene sets were further investigated (Table S4). Among these, one gene encoding cell cycle-related targets of E2F transcription factors, one gene up-regulated by activation of the PI3K-Akt-mTOR pathway and nine genes involved in the p53 pathway were up-regulated in the PEITC-treated Huh7.5.1 cells (Table 2). All these results showed that MAPK, PI3K-Akt and p53 signaling pathways could be activated by PEITC, suggesting the potential regulatory mechanisms of PEITC in inhibiting proliferation, migration and invasion, stimulating oxidative stress and inducing cell cycle arrest and apoptosis in Huh7.5.1 cells.

## DISCUSSION

HCC is one of the most common cancers in the world. Hepatectomy is the main feasible strategy for patients in the early stage (Fan et al., 2023). However, the advanced stage remains the most frequent presentation at diagnosis with limited effective treatment options, despite the improvement in the screening and diagnosis of HCC over the past years, and drug therapy is receiving increasing attention (Chen et al., 2020; Gordan et al., 2020). Isothiocyanates (ITCs) are one type of natural compounds exclusively

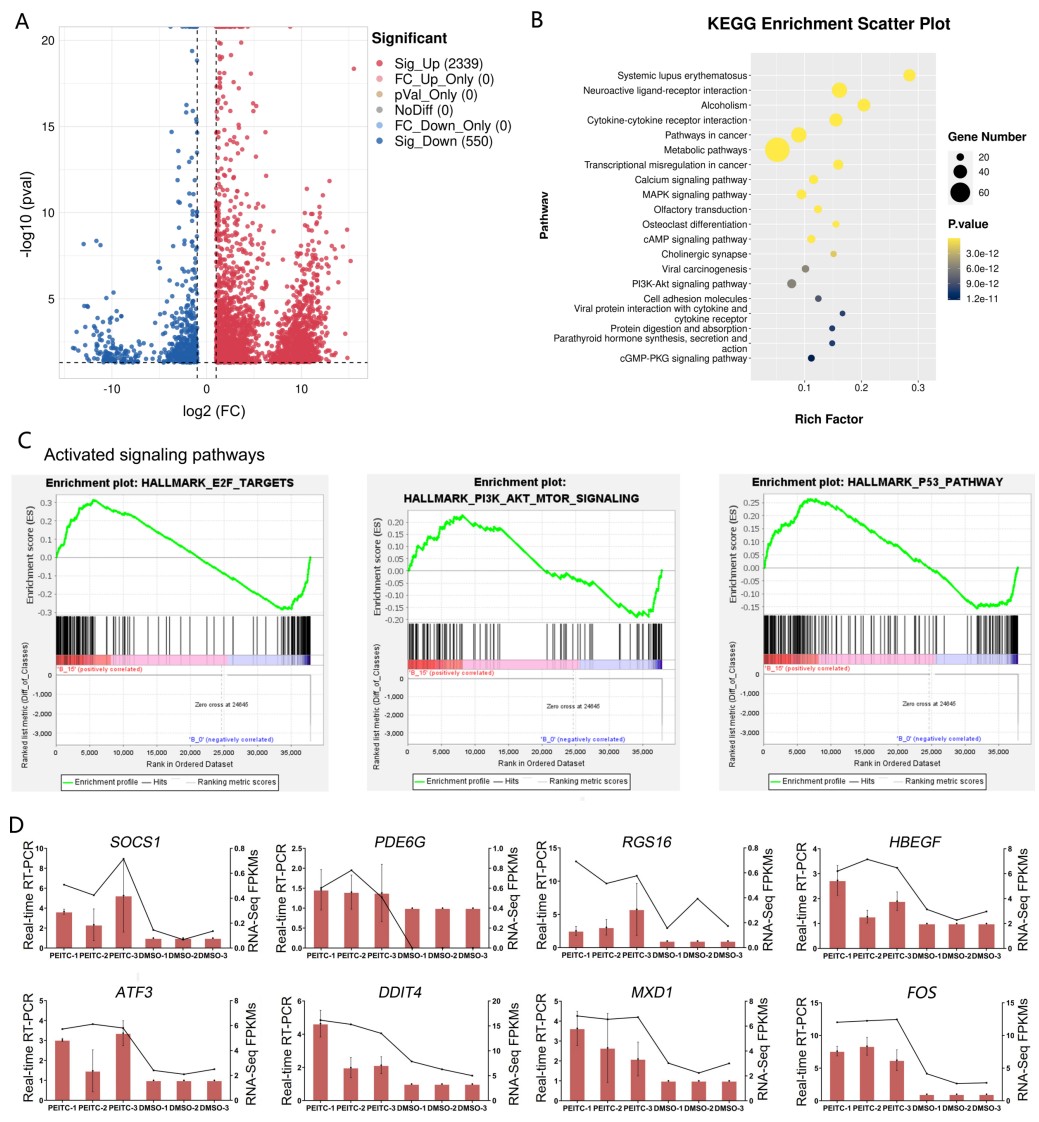

**Figure 6** **Transcriptome sequencing analysis of Huh7.5.1 cells incubated with or without PEITC.** (A) Volcano plot showing significantly differentially expressed genes (DEGs) with log2 fold change (FC) $\geq 1$ or $\leq -1$ and $p$-value $< 0.05$. (B) KEGG pathway enrichment analysis of DEGs with top 20 enrichment scores in Huh7.5.1 cells treated with PEITC. (C) Gene set enrichment analysis (GSEA) of the activated gene sets in the PEITC-induced Huh7.5.1 cells. Hallmarks E2F targets, PI3K-AKT signaling and p53 pathway were three PEITC associated activated gene sets in the PEITC-treated Huh7.5.1 cells. NES > 1.5, $p < 0.05$, $q < 0.25$. (D) Expression levels of *SOCS1*, *PDE6G*, *RGS16*, *HBEGF5*, *ATF3*, *DDIT4*, *FOS* and *MXD1* verified by quantitative real-time PCR in Huh7.5.1 cells incubated with or without PEITC. The columns indicate the relative mRNA levels of eight DEGs. The lines show the FPKM expression data of transcriptome sequencing.

abundant in cruciferous vegetables, which have potentially health-promoting activities (anti-oxidant, anti-microbial and anti-carcinogenic properties) (*Zhang et al., 2022*). Several ITCs, especially sulforaphane (SFN) and indole-3-carbinol (I3C), have been frequently assessed for their anti-carcinogenic effects (*Hajra et al., 2018*; *Mastuo et al., 2020*;

**Table 1  The gene sets significantly enriched in Huh7.5.1 cells after the treatment of PEITC based on GSEA.**

| Hallmark gene sets | ES | NES | NOM p-val | FDR q-val |
|---|---|---|---|---|
| HALLMARK_OXIDATIVE_PHOSPHORYLATION | 0.51 | 8.27 | 0.000 | 0.000 |
| HALLMARK_MYC_TARGETS_V1 | 0.44 | 7.38 | 0.000 | 0.000 |
| HALLMARK_ADIPOGENESIS | 0.37 | 6.06 | 0.000 | 0.000 |
| HALLMARK_DNA_REPAIR | 0.38 | 5.49 | 0.000 | 0.000 |
| HALLMARK_MTORC1_SIGNALING | 0.33 | 5.29 | 0.000 | 0.000 |
| HALLMARK_E2F_TARGETS | 0.31 | 5.16 | 0.000 | 0.000 |
| HALLMARK_UNFOLDED_PROTEIN_RESPONSE | 0.38 | 4.71 | 0.000 | 0.000 |
| HALLMARK_P53_PATHWAY | 0.26 | 4.25 | 0.000 | 0.000 |
| HALLMARK_GLYCOLYSIS | 0.25 | 4.22 | 0.000 | 0.000 |
| HALLMARK_HEME_METABOLISM | 0.26 | 4.10 | 0.000 | 0.000 |
| HALLMARK_FATTY_ACID_METABOLISM | 0.28 | 4.09 | 0.000 | 0.000 |
| HALLMARK_MITOTIC_SPINDLE | 0.25 | 4.07 | 0.000 | 0.000 |
| HALLMARK_UV_RESPONSE_UP | 0.25 | 3.71 | 0.000 | 0.000 |
| HALLMARK_PROTEIN_SECRETION | 0.30 | 3.60 | 0.000 | 0.000 |
| HALLMARK_XENOBIOTIC_METABOLISM | 0.21 | 3.47 | 0.000 | 0.000 |
| HALLMARK_PEROXISOME | 0.26 | 3.12 | 0.000 | 0.000 |
| HALLMARK_BILE_ACID_METABOLISM | 0.25 | 3.08 | 0.000 | 0.000 |
| HALLMARK_PI3K_AKT_MTOR_SIGNALING | 0.23 | 2.80 | 0.000 | 0.000 |
| HALLMARK_ESTROGEN_RESPONSE_LATE | 0.17 | 2.74 | 0.000 | 0.000 |
| HALLMARK_MYOGENESIS | 0.13 | 2.09 | 0.000 | 0.004 |
| HALLMARK_SPERMATOGENESIS | 0.15 | 1.90 | 0.004 | 0.011 |

**Notes.**
NES > 1.5, NOM $p$-val < 0.05, and FDR q-val < 0.25.

**Table 2  The differentially expressed core enrichment genes of MAPK, PI3K-Akt-mTOR and p53 signaling pathways in Huh7.5.1 cells after PEITC treatment.**

| Hallmark gene sets | Genes | Rank metric scores | Running ES | Core enrichment | log2 (Fold change(FC)) | p-value | Regulation |
|---|---|---|---|---|---|---|---|
| HALLMARK_PI3K_AKT_MTOR_SIGNALING | DDIT3 | 6.333 | 0.0209 | Yes | 1.476354886 | 1.56E−20 | up |
| | FOS | 9.063 | 0.0052 | Yes | 1.945767906 | 3.70E−39 | up |
| | DDIT4 | 8.61 | 0.01 | Yes | 1.2317004 | 4.21E−15 | up |
| | DDIT3 | 6.333 | 0.0375 | Yes | 1.476354886 | 1.56E−20 | up |
| | MXD1 | 3.94 | 0.0566 | Yes | 1.279879903 | 5.13E−25 | up |
| HALLMARK_P53_PATHWAY | HBEGF | 3.81 | 0.0659 | Yes | 1.240705667 | 2.16E−15 | up |
| | ATF3 | 3.55 | 0.0745 | Yes | 1.335222911 | 6.33E−26 | up |
| | PPM1D | 1.637 | 0.1324 | Yes | 1.053274531 | 4.25E−22 | up |
| | SOCS1 | 0.43 | 0.2233 | Yes | 2.240632636 | 0.000376018 | up |
| | RGS16 | 0.353 | 0.2539 | Yes | 1.303898481 | 0.010761383 | up |
| HALLMARK_E2F_TARGETS | PPM1D | 1.637 | 0.1566 | Yes | 1.053274531 | 4.25E−22 | up |

*Lv et al., 2020a*). The anti-oncogenic properties of PEITC, in terms of inhibiting the proliferation, migration and invasion and/or inducing cell apoptosis and cycle arrest, have also been demonstrated in various cancers, such as human breast, prostate, lung, cervical and bile duct cancers (*Xiao et al., 2005*; *Kassie et al., 2010*; *Tang et al., 2011*; *Huong et al., 2012*; *Tusskorn et al., 2013*; *Dai et al., 2016*; *Zhang et al., 2016*; *Soundararajan & Kim, 2018*). Especially in the field of HCC treatment, the liver's unique microenvironment, which possesses abundant blood supply and distinct metabolic characteristics, presents an ideal platform for the action of PEITC. More importantly, the chemical properties of PEITC enable it to exist stably within the liver's microenvironment and to be efficiently absorbed and utilized by liver cells, thereby enhancing its targeted attack on HCC cells (*Hecht, 1996*; *Gupta et al., 2014*; *Li & Zhang, 2017*; *Yang et al., 2024*). Therefore, PEITC has demonstrated significant potential in the treatment of HCC. Nevertheless, the full picture of the therapeutic effects and action mechanisms of PEITC on HCC has not yet been completely unveiled.

In the present study, we evaluated the effects of PEITC on human HCC cell line Huh7.5.1 and found that PEITC significantly inhibited the proliferation, induced apoptosis, promoted cell cycle arrest, and inhibited the invasion and metastasis of Huh7.5.1 cells, which were accompanied by the generation of ROS and a decline in MMP. Previously, PEITC has been demonstrated to significantly induce human hepatoma RLC/RPF/5 to undergo mitochondrial-related apoptosis, substantiated by the up-regulation of p53 and Bax protein, the down-regulation of Bcl-1, the cleavage of Bid, the release of cytochrome C, the activation of Caspase-3/8/9, the generation of ROS and the decrease of MMP (*Wu, Ng & Lin, 2005*). In addition, the effects of pro-apoptosis and cell cycle arrest and the generation of ROS induced by PEITC were reported in HepG2 cells (*Rose et al., 2003*). Another study in HepG2 cells also showed that PEITC could induce the ROS generation and affect microtubule depolymerization, leading to apoptotic and necrotic cancer cell death (*Pocasap, Weerapreeyakul & Thumanu, 2018*). In addition to its pro-apoptosis effect, an inhibitory role of PEITC and its N-acetylcysteine conjugate on cancer cell proliferation, adhesion, invasion, migration and metastasis was revealed in SK-Hep1 human hepatoma cells, mediated by the down-regulation of MMP and the up-regulation of TIMPs (*Hwang & Lee, 2006*). All these previous studies, including the present study, indicated that PEITC has anti-proliferation, anti-migration, pro-apoptosis and cycle arrest-promoting activities on HCC.

What is worth noting is the potential relationship between the decrease of MMP, the elevated ROS levels, and the DNA damage induced by PEITC treatment. There is an increasing amount of evidence indicating that ITCs can trigger a series of mitochondria-related apoptotic responses, including the decrease of MMP, the increase of Bax expression, and the decrease of Bcl-2 expression, ultimately promoting the enhancement of MOMP, which further causes the release of cytochrome C and promotes the production of ROS that causes DNA damage in HCC (*Zhang et al., 2022*). Moreover, ROS accumulation has been proved to cause DNA damage and apoptosis in various cancers (*Barzilai & Yamamoto, 2004*; *Guachalla & Rudolph, 2010*; *Guo et al., 2014*; *Sadiq, 2023*). Here, our results revealed the reduction of MMP, the generation of ROS, and DNA damage in the PEITC-treated

Huh7.5.1 cells. The activated gene sets associated with the p53 signaling and DNA repair based on GSEA also provided substantiation for the breakdown of DNA integrity. In conjunction with these results, our data propose that PEITC could induce DNA damage through the ROS generation triggered by the MMP decline in Huh7.5.1 cells, which requires further experimental verification.

During the complex multi-step process of cancer development, the balance between the activation and inactivation of tumor suppressor genes and proto-oncogenes plays a critical role in cancer progression (*Wang et al., 2002*; *Bishayee, 2014*; *Alqahtani et al., 2019*). After PEITC treatment, we found that the expression of genes (*FOS*, *HBEGF*, *RGS16*, *MXD1*, and *SOCS1*) involved in cell proliferation, differentiation, migration, invasion, apoptosis and genes (*ATF3* and *DDIT4*) associated with cell signal transduction and oxidative stress were significantly changed in Huh7.5.1 cells, confirming the anti-neoplastic effect of PETIC at the genetic level. Further transcriptome sequencing analysis revealed that the typical MAPK, PI3K-Akt and p53 signaling pathways were activated in the PEITC-treated Huh7.5.1 cells. Tumorigenesis has been associated with alterations in several classical signaling pathways (*Doerfler et al., 2001*). The involvement of multiple signaling pathways in aggressive cancers suggests that it may be necessary to target several pathways simultaneously for efficacy (*Liu et al., 2008*; *Li et al., 2012*). MAPK signaling pathways (including JNK/MAPK, ERK/MAPK, and p38/MAPK) are closely related to the occurrence and development of HCC (*Wu et al., 2020*; *Moon & Ro, 2021*). PEITC has been proven to target MAPK signaling pathways to exert its anti-cancer effect. It is capable of activating the JNK/MAPK signaling pathway in human colorectal cancer cells HT-29, thereby significantly inducing the G1 cell cycle arrest, and the JNK/MAPK and p38/MAPK signaling pathways in ovarian cancer cells OVCAR-3, which in turn activate Caspase-3, thereby facilitating the occurrence of apoptosis (*Wu et al., 2013*; *Islam et al., 2016*). Another study indicated that PEITC induces ferroptosis, autophagy, and apoptosis in K7M2 osteosarcoma cells by activating ROS-related MAPK signaling pathways (*Lv et al., 2020b*). In the present study, we observed that PEITC treatment led to cell cycle arrest at the S phase in Huh7.5.1 cells, which may be associated with the hyper-activation of the ERK/MAPK pathway. It has been demonstrated that the ERK/MAPK pathway plays a crucial role in regulating cell growth and differentiation, and its hyper-activation can trigger a cascade of intracellular responses, resulting in cell cycle arrest at the G1/S or G2/M checkpoints, thereby causing cell cycle blockage (*Kim & Choi, 2010*). Additionally, the activation of JNK/MAPK and p38/MAPK signaling pathways may intensify the cellular response to oxidative stress, promoting the accumulation of ROS, which in turn affects the normal physiological functions of the cell (*Wang et al., 2018*; *Kwak et al., 2023*). As a tumor suppressor protein, p53 plays a central role in how cells respond to DNA damage and stress (*Ou & Schumacher, 2018*; *Liebl & Hofmann, 2019*; *Vaddavalli & Schumacher, 2022*). Our transcriptome sequencing analysis revealed that treatment with PEITC significantly activated the p53 pathway, which in turn initiated the expression of downstream genes, including those that regulate cell cycle arrest and apoptosis (such as the *GADD45*, *FOS*, and *DDIT3* genes). This cascade of events ultimately led to an increase in MOMP, the release of cytochrome C, and the activation of members of the caspase kinase family, culminating in

the occurrence of apoptosis (*Carlsson et al., 2022*). The PI3K-Akt signaling pathway is often altered in a variety of cancer types, associated with cell survival, proliferation, and metabolic regulation (*Jiang et al., 2020*; *Su et al., 2023*). We found that the treatment of Huh7.5.1 cells with PEITC triggers the activation of the PI3K-Akt signaling pathway, which might be due to the pathway being temporarily activated as an emergency mechanism for cell survival when cells are under external stress, such as oxidative stress and DNA damage induced by PEITC, in an attempt to resist these pressures (*Chang et al., 2003*; *Wu et al., 2019*; *Huang et al., 2020*; *Jiang et al., 2020*; *Wang et al., 2022*; *Su et al., 2023*). However, when cellular damage exceeds the capacity for repair, the activation of the PI3K-Akt pathway fails to prevent the cells from proceeding to apoptosis. Instead, it may, through interactions with the MAPK and p53 signaling pathways, collectively promote the occurrence of apoptosis. In summary, PEITC intervenes in the biological behavior of Huh7.5.1 cells on multiple levels by synergistically activating signaling pathways such as MAPK, p53, and PI3K-Akt. There may be interactions among these pathways, jointly constructing a complex network. Their coordinated action ultimately leads to cell cycle arrest, oxidative stress, DNA damage, mitochondrial dysfunction, and apoptosis, thereby effectively inhibiting the growth, migration, and survival of Huh7.5.1 cells, achieving anti-cancer effects.

Although our study has revealed the activation of the MAPK, PI3K-Akt, and p53 signaling pathways in Huh7.5.1 cells after PEITC treatment, our understanding of its mediated anti-HCC mechanisms remains preliminary. In the future, we will proceed to unveil whether these pathways are directly activated by PEITC or whether they are secondary effects resulting from other cellular changes induced by PEITC, and elucidate how these signaling pathways interact within cells. In addition, we will focus on exploring other potential signaling pathways affected by PEITC, and assess the therapeutic efficacy of PEITC combined with extant treatment methods (such as targeted therapy, immunotherapy, and chemotherapy) (*Kassie & Knasmüller, 2000*; *Tsou et al., 2013*; *Denis et al., 2014*; *Ma et al., 2017*; *Shin et al., 2021*). Moreover, translating these significant effects from *in vitro* settings into clinical applications presents numerous challenges. Future research needs to delve into the therapeutic effectiveness and safety of PEITC *in vivo*, including validating its anti-tumor activity *via* preclinical animal models and conducting preliminary human clinical trials to assess its efficacy and safety in humans. Simultaneously, in view of the bioavailability and metabolic characteristics of PEITC, developing innovative and effective delivery systems, such as nanoscale fluoropyrimidine polymer CF10 (*Okechukwu et al., 2024*), is also a focus of future research to enhance therapeutic efficacy in HCC. To sum up, while our research provides promising preliminary data for the application of PEITC in the treatment of HCC, more studies are needed to fully understand its potential and limitations in clinical therapy. Through these efforts, we look forward to the development of more effective HCC treatment options with fewer side effects, offering better therapeutic choices for patients.

## CONCLUSIONS

The present study demonstrated that PEITC could suppress the proliferation, trigger mitochondrial-related apoptosis, induce cell cycle arrest and inhibit the invasion and

migration of HCC cells Huh7.5.1, which were accomplished by the decrease of MMP and the generation of intracellular ROS. The discovery of the inhibitory effects of PEITC on HCC cells Huh7.5.1, along with its role in modulating key signaling pathways, represents a significant breakthrough in the treatment of HCC. In-depth investigation into the underlying mechanisms not only enriches our understanding of PEITC's therapeutic potential but also facilitates its exploration as a promising therapeutic agent against HCC. These efforts may reveal novel targets for HCC treatment and offer a prospective therapeutic agent for improving the prognosis of patients with HCC.

## ACKNOWLEDGEMENTS

We thank Dr. Heng Dong (Hangzhou Normal University, China) for the critical comments and editing on this manuscript.

### Funding

This research was funded by the Basic Scientific Research Projects of Wenzhou [N20210003] and the Master's Innovation Foundation of Wenzhou University [3162023003045 and 3162024004093]. The funders had no role in study design, data collection and analysis, decision to publish, or preparation of the manuscript.

### Grant Disclosures

The following grant information was disclosed by the authors:
The Basic Scientific Research Projects of Wenzhou: N20210003.
The Master's Innovation Foundation of Wenzhou University: 3162023003045, 3162024004093.

### Competing Interests

The authors declare there are no competing interests.

### Author Contributions

- Jiao Du performed the experiments, analyzed the data, prepared figures and/or tables, authored or reviewed drafts of the article, and approved the final draft.
- Yuting Zhang performed the experiments, analyzed the data, prepared figures and/or tables, authored or reviewed drafts of the article, and approved the final draft.
- Jiajia Chen performed the experiments, analyzed the data, prepared figures and/or tables, and approved the final draft.
- Libo Jin analyzed the data, prepared figures and/or tables, and approved the final draft.
- Liying Pan analyzed the data, prepared figures and/or tables, and approved the final draft.
- Pengyu Lei analyzed the data, prepared figures and/or tables, and approved the final draft.
- Sue Lin conceived and designed the experiments, authored or reviewed drafts of the article, and approved the final draft.

## DNA Deposition

The following information was supplied regarding the deposition of DNA sequences:

The RNA sequences of Huh7.5.1 cells with PEITC or DMSO treatment described here are available at SRA BioProject: PRJNA1072050.

## Data Availability

The raw data is available at Figshare: Jiao, Du (2024). Phenethyl isothiocyanate inhibits the carcinogenic properties of hepatocellular carcinoma Huh7.5.1 cells by activating MAPK/PI3K-Akt/p53 signaling pathways. figshare. Journal contribution. https://doi.org/10.6084/m9.figshare.25183820.v2.

## Supplemental Information

Supplemental information for this article can be found online at http://dx.doi.org/10.7717/peerj.17532#supplemental-information.

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
