# Peer review of "Phenethyl isothiocyanate inhibits the carcinogenic properties of hepatocellular carcinoma Huh7.5.1 cells by activating MAPK/PI3K-Akt/p53 signaling pathways"

_PeerJ, doi:10.7717/peerj.17532_

## Round 0.1 · original submission · Major Revisions

Your manuscript has been assessed by three independent reviewers. While there is an interest in the study, major concerns have been raised, including the need for more data and analysis, and clarification of some of the presented results, which should be fully addressed before we can move forward.

Reviewer 1 ·

Basic reporting

no comment

Experimental design

no comment

Validity of the findings

no comment

Additional comments

Abstract:
Clarity and Conciseness: The abstract provides a clear overview of the study's objectives, methods, results, and conclusion. However, it could be more concise in some areas. For example, the sentence "Accumulating evidence demonstrated that PEITC exerts anticancer properties in multiple cancers involving the lung, the bile duct and the prostate" could be shortened to "PEITC has demonstrated anticancer properties in various cancers, including lung, bile duct, and prostate cancers."
Specificity: The abstract mentions that PEITC has "mysterious inhibitory roles" in HepG2 and Sk-Hep-1 cells. It would be better to specify what these roles are or to rephrase this to avoid ambiguity.

Introduction:
Clarity and Coherence: The introduction provides a clear and coherent overview of the background and significance of the study. However, it could be more concise in some areas. For example, the discussion on the limitations of targeted medication in line 66 could be shortened.
Research Objectives: The objectives of the study are clearly stated in lines 105-107. It would be helpful to briefly mention the expected outcomes or hypotheses based on these objectives.
Consistency in Terminology: Ensure consistency in the use of terminology throughout the introduction. For example, "HCC" is used interchangeably with "primary liver cancer" and "hepatocellular carcinoma." It would be better to consistently use "HCC" after the first full mention of "hepatocellular carcinoma."
Overall, the introduction provides a solid foundation for the study. Addressing the above comments could enhance its clarity and impact.
Materials and Methods:
Cell Culture and Reagents: The description of cell culture conditions and reagents used is clear and detailed. However, it would be helpful to include the passage number range of the Huh7.5.1 cells used in the experiments to ensure reproducibility.
qRT-PCR: The qRT-PCR protocol is well described, but it is essential to include information about the primer sequences used for each gene of interest. This will allow other researchers to replicate the experiments accurately.
Reproducibility and Rigor: To enhance the reproducibility of the study, it is crucial to provide detailed information about the number of biological and technical replicates for each experiment. Furthermore, including any measures taken to minimize bias (e.g., randomization, blinding) would increase the rigor of the study.

Results:
Antiproliferative Effects of PEITC: The results showing the dose-dependent decrease in cell viability and colony formation are clear and well-presented. However, it would be beneficial to include statistical significance values (e.g., p-values) in the figures to support the claims made in the text.
Apoptosis Induction: The findings indicating that PEITC induces apoptosis in Huh7.5.1 cells are supported by multiple assays, including Annexin V-FITC/PI staining, MMP measurement, and Caspase-3 activity detection. The use of multiple assays strengthens the conclusion that PEITC induces apoptosis.
Oxidative Stress and DNA Damage: The results demonstrating increased ROS levels and DNA damage in PEITC-treated cells are intriguing. However, it would be interesting to explore whether the observed DNA damage is a direct effect of ROS or if other mechanisms are involved.
Inhibition of Migration and Invasion: The wound healing and Transwell invasion assays show that PEITC inhibits the migratory and invasive abilities of Huh7.5.1 cells. Including quantification of the results (e.g., percentage of wound closure, number of invaded cells) would enhance the interpretation of these findings.
Activation of Signaling Pathways: The activation of MAPK, PI3K-Akt, and p53 signaling pathways by PEITC is a significant finding. It would be interesting to investigate whether these pathways are directly activated by PEITC or if they are secondary effects resulting from other cellular changes induced by PEITC.
Overall, the Results section provides compelling evidence for the anticancer effects of PEITC on Huh7.5.1 cells. Addressing the above comments could further strengthen the findings and their interpretation.

Discussion:
Comprehensive Review of PEITC's Anticancer Effects: The discussion provides a thorough overview of the anticancer properties of PEITC in various cancer types, including HCC. It would be beneficial to include a brief comparison of PEITC's efficacy in HCC compared to other cancers to highlight its potential as a therapeutic agent specifically for HCC.
Mechanisms of Action: The discussion on the activation of MAPK, PI3K-Akt, and p53 signaling pathways by PEITC is well-presented. However, it would be valuable to provide more detailed explanations of how these pathways contribute to the observed anticancer effects in Huh7.5.1 cells.
Future Directions: The discussion briefly mentions that many signaling pathways are involved in PEITC-mediated HCC therapy. It would be beneficial to elaborate on potential future research directions, such as exploring other signaling pathways or combining PEITC with other therapeutic agents.
Clinical Relevance: The discussion emphasizes the need for new chemotherapy drugs with high bioavailability and minimal adverse effects. A brief mention of the potential clinical implications of this study's findings and the steps needed to translate these findings into clinical practice would be valuable.

Reviewer 2 ·

Basic reporting

Du et al. found that PEITC exhibits anti-proliferative and pro-apoptosis effects in one HCC cell line. The RNA-seq results indicate that MAPK/PI3K-Akt/p53 signaling pathways may be involved in PEITC-mediated anticarcinogenic activities. However, the current results cannot support the authors' conclusion.

Experimental design

The most crucial issue is that all results provided by authors are the effects of PEITC on cells; however, 1)the inner mechanism is mainly unrevealed, so further investigation is urgently needed; 2) in vivo data is required.

Validity of the findings

The other key issue is the presentation of the error bars: 1) in Figures 1, 2, and 3, the error bars have obvious gaps with the related column; authors should explain this; 2) there are several formats of the figures, some of the bar charts contain the dots(Figure 5) while some do not(Figure 1, 2, 3, and 6).

·

Basic reporting

The article should include a sufficient introduction and background to demonstrate how the work fits into the broader field of knowledge.

Experimental design

All the experimental protocols are well-defined. However, some queries were raised in this regard that has been mentioned in the attached file.

Validity of the findings

The experiments were replicated thrice and are statistically sound. Conclusions are well stated and linked to research questions. However, ambiguity prevails in the research question that has been stated in the attached comments.

Additional comments

The authors of this research article tried to explore potential of Phenethyl isothiocyanate to inhibit the carcinogenic properties of Huh7.5.1 cells. The work done is extensive and explanatory. However, some inadequacy was noted in the manuscript which needs to be addressed for the improvement of the quality of the research article considering the publication criteria of the highly esteemed journal like PeerJ. The comments are given below:


1. It is not clear how Huh 7.5.1 cells are different from that of HepG2 and SkHep-1 cells. The authors should justify the same in the Introduction part.
2. Did the author check for CD81, a plasma membrane protein essential for viral replication?
3. The authors didn’t check the effect of PEITC in normal cell line.
4. Why the authors have kept the incubation time for 48 hours?
5. Identification and characterization of genomic markers in human hepatoma cells have already been done (Ref: Kawamoto M, Yamaji T, Saito K, Shirasago Y, Satomura K, Endo T, Fukasawa M, Hanada K, Osada N. Identification of Characteristic Genomic Markers in Human Hepatoma HuH-7 and Huh7.5.1-8 Cell Lines. Front Genet. 2020 Oct 9;11:546106. doi: 10.3389/fgene.2020.546106. PMID: 33193621; PMCID: PMC7581915). What are the novel findings of this paper is not clear?
6. The authors should mention the p53 status of Huh 7.5.1 cells at basal levels.
7. It is not clear from the western blot results that how come when IκBα is high then NFκBp65 is also high because IκBα is supposed to inhibit the expression and activation of NFκB.
8. IC50 of PEITC in Huh 7.5.1 cells is 29.61 μM. Then the basis for selecting 15 and 30 μM is not clear.
9. Some grammatical errors are noted which need rectification.

Considering these issues in my opinion the research article needs revision for better precision and interpretation.

---

## Round 0.2 · Minor Revisions

All reviewers and the editor agree that the revisions have significantly improved the manuscript. Reviewer 3 still has a few comments on typos and others, which I encourage the authors to address.

Reviewer 1 ·

Basic reporting

Introduction or Background Section:
When setting the stage for discussing innovative cancer treatments or the challenge of treating metastatic liver cancer, citing PMID: 38611037 study could help underscore the necessity and potential of new technologies in overcoming these challenges.

Specifically, in the segment where future research directions are contemplated, particularly the part discussing the development of effective drug delivery systems to enhance the bioavailability and therapeutic impact of treatments in hepatocellular carcinoma (HCC).

The authors might consider discussing the role of innovative drug delivery systems in enhancing therapeutic efficacy in hepatocellular carcinoma (HCC). For example, the work by Okechukwu et al. (2024), which demonstrates significant improvements in treatment outcomes for liver metastasis through the use of nanoscale fluoropyrimidine polymer CF10, could serve as a valuable reference. This study underscores the potential of nanotechnology in enhancing drug efficacy and targeting, which could be relevant to the authors' investigations into similar strategies for PEITC

Experimental design

The reference, PMID: 37557926, seems to be important to support the current manuscript as it investigates benzofuran piperazine derivatives, highlights the ongoing efforts to synthesize and evaluate novel compounds that could potentially offer enhanced anticancer activity. Such efforts underscore the importance of chemical innovation in developing more effective and targeted therapies for conditions such as hepatocellular carcinoma. Additionally, this reference also validates the current version of survival assyays (MTT and Apoptosis assay). Authors may use this reference to support their mansucript.

Validity of the findings

no comment

Additional comments

no comment

·

Basic reporting

No Comment

Experimental design

No Comments

Validity of the findings

No Comments

Additional comments

1. INTRODUCTION: Line number # 110: The term ‘DRUG’ should be omitted as PEITC is not a drug.

2. Still some typo-errors and grammatical changes persist, which need rectification.

---

## Round 0.3 · accepted · Accept

The manuscript is now ready for publication.